# Sense of Coherence and Self-Rated Aggression of Adolescents during the First Wave of the COVID-19 Pandemic, with a Focus on the Effects of Animal Assisted Activities

**DOI:** 10.3390/ijerph20010769

**Published:** 2022-12-31

**Authors:** Ákos Levente Tóth, Zsuzsanna Kívés, Etelka Szovák, Réka Kresák, Sára Jeges, Bendegúz Kertai, Imre Zoltán Pelyva

**Affiliations:** 1Institute of Sport Sciences and Physical Education, Faculty of Science, University of Pécs, 7624 Pécs, Hungary; 2Institute for Health Insurance, Faculty of Health Sciences, University of Pécs, 7621 Pécs, Hungary; 3Doctoral School of Health Sciences, Faculty of Health Sciences, University of Pécs, 7621 Pécs, Hungary; 4Institute of Psychology, Faculty of Humanities and Social Sciences, University of Szeged, 6722 Szeged, Hungary

**Keywords:** COVID-19 pandemic, sense of coherence, aggression, animal assisted activities

## Abstract

The COVID-19 pandemic has caused extreme deviations from everyday life. The aim of this study was to investigate how these deviations affected adolescents’ sense of coherence and their level of aggression, and whether this was influenced by their relationship with animals, especially horses. In two random samples of students from vocational schools in Hungary, taken in June 2018 and June 2020 (n1 = 525, n2 = 412), separate groups were drawn from those who had regularly engaged in equine-assisted activities (ES) and those who had not (OS) before the pandemic. Data were collected using an anonymous, paper-based questionnaire, and during the pandemic an online version of the Sense of Coherence (SOC13) and Bryant–Smith (B12) scales. During the pandemic, boys’ sense of coherence weakened and their aggressiveness increased. Multiple linear regression analyses showed that, regardless of gender and age group, increased time spent using the internet (*p* < 0.001), a lack of classmates (*p* = 0.017), reduced time spent outdoors (*p* = 0.026) and reduced physical activity (*p* < 0.038) during the pandemic significantly increased the tendency for aggressive behavior, whereas being with a horse or pet was beneficial (*p* < 0.001). The changes imposed by the curfew were rated as bad by 90% of the pupils, however, those with a strong sense of coherence felt less negatively about them. Schools should place a great emphasis on strengthening the students’ sense of coherence.

## 1. Introduction

To control the COVID-19 outbreak, restrictive measures have been introduced around the world, including in Hungary. During the first wave of the epidemic, the lockdown was implemented from March to June 2020 in Hungary [1]. These restrictions also affected the everyday lives of adolescents. Education changed to an online format from one day to the next, and face-to-face encounters with other students were abolished. The daily news of the growing epidemic and the increase in the number of deaths, which also manifested in their immediate environment, were traumatizing for adolescents. A scoping review of 11 studies regarding the effects of COVID-19 on the lives of adolescents showed that the pandemic can be considered a determinant that affects different dimensions of adolescents’ lives [2]. This result was reinforced by other studies on the diverse psychological symptoms and behavioral changes in this age group during the lockdown [3,4]. At-risk groups, like justice-involved youth, also showed increased distress and antisocial behaviors [5].

Comparing the levels of anxiety between different age groups, researchers found that children and adolescents were more apt to feel depression and anxiety during and after the pandemic. Social distancing, school closures and isolation caused disturbances in sleep and appetite, as well as impairment in social interactions, which may even have long-term adverse consequences on mental health. This tendency highlights the need for effective mental health strategies focusing on the needs of children and adolescents [6,7,8], even more so as disadvantaged youth groups, e.g., those with a lower level of education, might develop negative coping styles found to be strong predictors of youth mental health [9]. Prolonged confinement during the COVID-19 pandemic triggered negative emotional reactions among adolescents. Besides anxiety, children and adolescents also experienced anger, sadness, and boredom/emptiness, against which living in a rural area was found to be a protective factor [10], while low socioeconomic status and limited living space were found to be aggravating factors [11].

The research into adolescent youths’ sense of coherence carries a long history [12,13,14,15,16,17,18] and has gained even more relevance during the COVID-19 pandemic.

Aaron Antonovsky established the concept of a sense of coherence (SOC) in 1979 to explain why some people fall ill under stress while others stay healthy. The SOC consists of three main components: comprehensibility, manageability, and meaningfulness. Comprehensibility, a cognitive aspect, refers to the extent to which one might perceive both internal and external stimuli as being understandable in some kind of rational way [19]. This might also be related to the ability to see things as orderly, coherent, clear and structured. The idea of being able to create something structured out of a chaotic situation renders it much easier for us to understand the context in which we might weigh things or view life. Manageability, a behavioral aspect, bears a link to the degree to which we might be confident that we have resources at our disposal. These resources might then be used to help manage the stimuli that we are incessantly bombarded with [19]. Manageability also has to do with our ability to cope and solve problems, as well as our readiness to invest our time and energy into solving those problems—in other words, to manage those problems and comprehend them as a challenge rather than a burden. Meaningfulness, a motivational aspect, has to do with the extent to which we perceive our lives as having some kind of emotional meaning. This may also come into play when facing some sort of problem or challenge [19].

SOC is closely related to quality of life, preventive and health-promoting behaviors, and the ability to cope with stress and illness and adapt to difficult situations [20]. It also bears an outstanding role as a coping resource against stress and in adolescents’ mental health in general [21]. The preventive role of SOC against the stressor experience also helps to improve adolescents’ life satisfaction [22]. SOC was found to be a predictor of health-related quality of life and to positively impact health outcomes, thus helping youth better adapt to stressful events such as a pandemic [23]. Reduced physical activity and prolonged screen time negatively influenced health-related quality of life during the pandemic [24], further strengthening the importance of SOC. Cognitive reappraisal and humor also proved to be possible protective factors against the adverse effects of the COVID-19 pandemic and other prolonged stressors influencing mental health [25].

During periods of prolonged restriction, it can be assumed that the number of aggressive acts increases, as confirmed by a follow-up study among adults in the USA [26]. Research revealed that during the COVID-19 pandemic, family economic strain significantly worsened adolescent aggression through inter-parental and parent–child conflict [27].

Complex relationships exist between the proximity of pets, pet ownership, and the effects of confinement, which may be important to study when dealing with similar crisis situations [28]. Companion animals were found to be an important source of emotional support during the COVID-19 lockdown. They helped to maintain the owners’ mental health, preventing an increased feeling of loneliness [29]. The strength of human–animal bonds may serve as an indicator of mental health-related vulnerability, regardless of the animal species, and companion animals may act as social buffers for psychological distress and loneliness [29].

A study including almost 300 adolescents living in rural areas found that pet owners felt significantly less lonely than non-pet owners and that the extent of companion animal attachment was positively correlated to social support and inversely proportional to loneliness, suggesting that pet bonding interventions may reduce adolescents’ loneliness [30].

Vocational schools, where practical, field-based education plays a major role, have been particularly affected by the suspension or restriction of these activities. In those institutions where animal-assisted activities are also taught (e.g., horsemanship training), full restrictions were not feasible. Animals had to be fed and looked after, and pupils had to take part in the work. Physical activity has been correlated with and recommended to support children’s and adolescents’ psychological health while under the influence of COVID-19 [31]. This way, mandatory physical work might have had a positive effect on the quality of life these young people experienced.

Although not commonly thought of as pets, horses may also have a beneficial effect on psychosocial factors through positive relational, affectionate behavior, especially for boys, as within the horse–human connection they can learn how to display positive rather than damaging masculine behavior [32].

Equine-assisted activities are recommended approaches to alleviate mental and physical difficulties and enhance the health and well-being of at-risk humans [33,34], “The human-animal interaction (HAI) is associated with health. The health benefits demonstrated typically include reductions in depression and loneliness while enhancing social interaction or social skills, and decreasing anxiety and arousal. Other health benefits associated with companion animals include the promotion of exercise or physical activity.” [35].

Our study aimed (1) to describe the sense of coherence and trait aggressiveness in the target population before and during the first wave of the pandemic, and (2) to investigate what factors influenced the perception of changes in their daily lives caused by the lockdown, with a focus on human-animal interactions and animal-assisted activities.

## 2. Materials and Methods

### 2.1. Study Population

Our target group included 14–18-year-old students from 10 agricultural secondary schools in Hungary. They all took part in a four-year horse groom training program. These students had no diagnosed physical or psychological difficulties. As part of the schools’ curriculum, they spent two days (that is 9 to 13 h) per week with horses before the outbreak of the COVID-19 pandemic. On these occasions, they fed and groomed the horses, cleaned the stable, worked with the horses on the lunge, from the saddle, and practiced carriage driving. Later, these students are referred to as equine students (ES). Members of the control group consisted of students from the same schools who studied non-horse related, agricultural, or food industry vocations (e.g., gardening, animal husbandry, meat processing, baking) and thus, did not take part in any activities involving horses. They make up the other students (OS) group of the study.

A questionnaire survey was conducted at two points in time: before the pandemic (in 2018) and in May 2020, during the first wave of the pandemic (during the epidemic’s first wave, the lockdown was implemented from March to June 2020 in Hungary), when the curfew was still in place due to the COVID-19 epidemic and education was provided through online training. The data and results of the first survey serve as a control for testing the pandemic’s impact, as the samples do not differ significantly in terms of basic sociodemographic factors.

### 2.2. Methods

This study was a comparative cross-sectional investigation. Both surveys were anonymous questionnaires, including, among other items, the 13-item Sense of Coherence Questionnaire (SOC13) and the 12-item Aggression Questionnaire (B12) [36], both validated on the Hungarian population. The second survey also contained questions related to the pandemic, which concerned the students’ lifestyles, their changed circumstances and their perception of the impact of these.

In both surveys, the students were divided into two groups according to their vocational training, equine students (ES) and the other students (OS). During the pandemic, the ES group had the usual horse-assisted activities significantly restricted during the curfew, and the students were only minimally involved in horse care, feeding, watering, etc.

The original aim of the 2018 study (hereafter Research I) (N = 525) was to investigate the impact of horse-assisted activities on behavioral problems and prosocial behavior [37]. The present study will process and evaluate the sense of coherence (based on SOC13) and aggression (based on B12) scores of the study population before and during the pandemic.

Conducted online, the 2020 survey (hereafter Research II) used a google form to produce a digital version of the questionnaires (N = 412). Out of the total 550 children contacted, 412 completed the questionnaire and participated in the research. The response frequency was, therefore, 74.9%. For the statistical analyses, the number of sampled respondents (assuming 80% power and 5% type I error) based on our prior knowledge of the variables, and the estimation obtained with the G*Power software (version 3.1) is sufficient.

#### 2.2.1. Measurements

##### Sense of Coherence

Sense of coherence was measured with the Hungarian version of the 13-item Sense of Coherence Scale developed from the original 29 items, elaborated by Antonovsky [38] and validated on a Hungarian population [39]. A 7-point rating scale (mostly ranging from 1 = very rarely to 7 = very often) was used to measure regularity, where experienced feelings measure each item. “It often happens” indicated a weaker sense of coherence, while a higher total score would correspond to a stronger sense of coherence. The 13-item SOC scale contains the following items:

Five items referred to comprehensibility, e.g., “Do you have very mixed-up feelings and ideas? Very often (1) to very seldom or never (7)”, and “Does it happen that you have feelings inside you would rather not feel? Very often (1) to very seldom or never (7)”.

Four items referred to manageability, e.g., “Do you have the feeling that you’re being treated unfairly? Very often (1) to very seldom or never (7)” and “When something unpleasant happened in the past, your tendency was: to beat yourself up about it (1), to say ok, that’s that, I have to live with it and go on (7)”.

Four items referred to meaningfulness, e.g., “When you think about your life, you very often: feel how good it is to be alive (1), to ask yourself why you exist at all (7)” and “Doing the things you do every day is: a source of deep pleasure and satisfaction (1) to a source of pain and boredom (7)”, (in the case of this item, when calculating the score sum, the coding direction was inverted (1 = 7, 2 = 6, 7 = 1), so that answers suggesting a higher sense of coherence were scored higher).

Internal reliability indexed with Cronbach’s α was 0.769 in the first survey and 0.788 in the second, which was within the acceptable scope [40].

##### Trait Aggression

We used the Hungarian version [41] of Bryant–Smith’s 12-item aggression scale [36] to measure aggression. This is a self-report questionnaire assessing aggressive tendencies [42]. The validity of the scale for the Hungarian population was previously checked (Cronbach’s alpha for the total test was 0.84). It comprises four components of aggression: physical aggression, anger, hostility and verbal aggression. It contains 12 items (e.g., from the four subscales: “Given enough provocation, I may hit another person.” (physical), “I flare up quickly but get over it quickly.” (anger), “At times I feel I have gotten a raw deal out of life.” (hostility), “My friends say that I’m somewhat argumentative.” (verbal) rated on a 5-point Likert scale from 1 (Extremely uncharacteristic of me) to 5 (Extremely characteristic of me).

#### 2.2.2. Outcome Variables

The outcome variables are the 13 items of Sense of Coherence (SOC13) and its 3 elements, and the 12 items of the Bryant-Smith (B12) test and its 4 elements.

In the second survey, students’ overall perception of the pandemic was measured by the question “Overall, how would you rate your life under curfew compared to before the pandemic?”, which was answered on a five-point Likert scale (1 much worse, 2 worse, 3 neither worse nor better, 4 better, 5 much better).

#### 2.2.3. Explanatory Variables

In accordance with the goal of our research, we compare (similarly to the previous survey) students who regularly deal with horses (equine students—ES) to other students (OS) who have no connection to horses. This way, the variable “Do you deal with horses on a regular basis?” can take two values: yes or no. Another exploratory variable was also introduced, namely, whether students are closer to the beginning or the end of their studies, i.e., whether they have been working with horses for 1 year or 2–3 years, either in or out of school. In addition to the horse-assisted activity, we also looked at whether pupils had a pet (e.g., dog, cat, rabbit, or another small animal) and how much time they usually spent with them.

In the second study, we also considered the sense of coherence as an additional explanatory variable for the self-perception of the pandemic.

#### 2.2.4. Control Variable

Since, usually, a difference in the sense of coherence and aggressiveness exists between males and females [43,44], analyses were conducted separately for boys and girls. In the multivariate analyses, gender was used as a control variable.

### 2.3. Statistical Analysis

Statistical analyses were conducted, using SPSS v. 25. (IBM Corp. Released 2017. IBM SPSS Statistics for Windows, Version 25.0, IBM Corp., Armonk, NY, USA). The qualitative sample characteristics of the two surveys are compared using the chi2 test. Output variables are characterized by median and IQR, and mean and standard deviation. The Mann–Whitney test is used to compare variables with ranks, and the independent samples *t*-test is used for variables considered to be normally distributed. Binary logistic regression analysis and linear regression analysis are used for multivariate analysis.

## 3. Results

### 3.1. Study Population

Table 1 presents the composition of the samples in terms of gender, age group, and type of education (OS vs. ES).

According to age, gender, place of residence and type of education as possible influencing factors, the two samples do not differ significantly (Table 1).

### 3.2. Characteristics of Pupils’ Lives during the Epidemic Curfew

Before the outbreak, 43.3% lived in a student hostel. They were generally at their place of residence during the curfew. During this period, 5.3% (22 people) did not leave their homes at all; 48.5% (143 people) left their homes every 2–3 days or less; 41.2% (17 people) left their homes once or more times a day. Compared to before, 22.7% (56 people) spent about the same amount of time outdoors, 40.1% (165 people) less, and 37.2% (92 people) more. In terms of physical activity, the proportion of people who were less active than before also registered high, at 53.1% (131 people), but 21.9% were more active. No significant difference manifested between boys and girls in this respect. In their free time, a very high proportion, 74.1% (183) spent time with a pet (dog, cat, rabbit, guinea pig)—a significantly higher proportion of girls did so. Moreover, 40.8% (168) were involved with horses specifically, with the same proportion of boys and girls.

Education was delivered online; 93.0% of students used a mobile phone and 19.7% used a mobile phone exclusively, while the rest used a laptop, desktop, or tablet in addition to their mobile phone. On average, they spent 6.4 ± 2.5 h per day online, of which 4.2 ± 2.5 h were spent on compulsory school material. In comparison, the pre-pandemic survey found that respondents spent 5.4 ± 5 h per day on the internet. There is a very wide variation in this respect. During the pandemic, girls on average spent one hour more on the internet than boys, but the extra time was due to increased time spent on studying. The difference between boys and girls was significant (*p* < 0.050) in terms of time spent studying and time spent on the internet outside of school. Compared to before the epidemic, both boys and girls spent on average one hour more on the internet, also a significant difference.

### 3.3. Presentation of the Main Outcome Variables in Studies before and during the Pandemic

The scores on the subscales that make up the SOC and Bryant scales are presented as dichotomous variables along the median for the two studies, broken down by gender (Table 2).

Table 2 shows that on the SOC13 scale in the pre-epidemic sample, boys have a stronger sense of coherence than girls, while in the post-epidemic sample, this difference disappears. According to the Bryant12 test, the level of aggression in the first sample does not differ between boys and girls, but during the epidemic, the tendency towards aggression remains significantly stronger in boys in terms of hostility and verbal aggression, and the difference is also shown on the aggregated scale. Comparing the two data sets, it can be concluded that the boys’ sense of coherence was weakened, while aggression, in all its dimensions, was strengthened. For girls, this was only evident in physical aggression.

Using the method developed by Templeton [45] for the total scores of the SOC and aggressiveness questionnaire, the variables were quasi-normally distributed (Kolmogorov–Smirnov test *p* > 0.200). Statistical characteristics of the samples by gender are shown in Table 3.

### 3.4. Correlation between SOC and Aggressiveness

In both samples, a highly significant correlation exists between the total score on the SOC and the aggressiveness tests, with Pearson’s rho = 0.419 and −0.546 (*p* < 0.001). Based on Spearman’s correlation coefficient value, the components of SOC and each dimension of the aggression test are significantly correlated (*p* < 0.010).

In other words, the stronger the student’s sense of coherence, the lower the score on the trait aggressiveness test.

### 3.5. Comparison of Main Outcome Variables by Type of Education

The effect of animal-assisted activity on the sense of coherence and the propensity towards aggression was investigated using a multiple linear regression model. The dependent variable in the model was the SOC13 total score and the Bryant test’s total score, and the main explanatory variable was the response to the question of whether the participant had engaged in animal-assisted activities for a considerable time (ES vs. OS). Age was included as an independent variable in the model with two categories: aged 14–15 and aged 16–18. This variable indicates whether the respondent is at the beginning of vocational training or has been in training for 1–2 years. Gender was taken as a control variable.

According to the data from the first survey, the sense of coherence of the students who engaged in horse or animal-assisted activities was slightly stronger than that of the students in the control group (*p* = 0.048). The sense of coherence of boys was also significantly stronger than that of girls (*p* = 0.026). According to the survey conducted during the pandemic, the differences were not significant (Table 4).

Gender and age were not significant in either of the surveys; however, during the pandemic, the tendency toward aggression was significantly greater in the equine group (*p* = 0.027) (Table 5).

### 3.6. Comparison of the Overall Score of Aggressiveness by Sense of Coherence

According to both studies, the propensity to be aggressive is strongly influenced by manageability and meaningfulness. In the first study, comprehensibility was not significant, in the second study it was, but not as strongly (*p* = 0.044) as the other two components of SOC, which had a significance level of *p* < 0.001. In the second study, the stronger tendency for aggressiveness regarding those in the equestrian group was found to be a significant factor even when controlling for the sense of coherence (Table 6).

### 3.7. Impact of Changes during Curfew on Aggressiveness (Bryant 12)

As shown above, especially among boys, the scores of the Bryant test were higher in the sample taken during the epidemic than in the sample taken before the epidemic. We next sought to determine whether changes in pupils’ lifestyles during the curfew period might be associated with this result. We looked at the following factors: whether they lived in a town or a village at the time of the curfew, how often they were able to leave home, how physically active they were compared to before, how much time they spent outdoors, whether they spent more or less time with horses or pets, how much time they spent on the internet during compulsory lessons or studying and how much time they spent for leisure, whether they missed their classmates or friends, and whether they had the opportunity to meet them in person.

In Table 7, only the standardized regression coefficients of the significant factors and the results of their significance tests are shown, as well as the control variables.

It can be seen that, regardless of gender and age group, increased time spent on the internet, less time spent outdoors, less physical activity and lack of classmates increase the tendency to be aggressive, while spending more time with horses or pets is beneficial.

### 3.8. Pupils’ Overall Rating on the Impact of the Curfew on Their Lives

The breakdown of responses to the question “Overall, how would you rate your life under curfew compared to before the epidemic?” by gender is shown in Figure 1.

According to the chi2 test, the distribution of responses between genders did not differ significantly (*p* = 0.234); 49.5% of students rated their life under curfew as worse. We examined the role of the student’s sense of coherence and their tendency toward aggressiveness in this rating. The result obtained with the multiple logistic regression model showed that only the component of comprehensibility in the sense of coherence was significant for the rating “worse”. The stronger the comprehensibility component was, the more likely the respondent was to rate his life under curfew as the same or better.

The questionnaire also asked the student to justify their answer to the above question in the form of an open question. Due to the relatively few responses (19% of respondents gave reasons), only the most frequent answers are highlighted, without statistical analysis. Positive comments were “I could sleep longer”, “I could manage my time myself”, “I could spend more time with my pets”, “I could spend more time with my family”, and “I didn’t have to go to school and I could work and exercise”. The most common negative comment was “I was locked up”.

## 4. Discussion

The results of a comparative study on adolescents in vocational schools before the COVID-19 pandemic and during the curfew in June 2020 confirm that adolescents have negative experiences of social distancing and school closures. Specifically, these measures may promote violent situations or aggressive behavior in the home environment [2].

This is indicated by the fact that scores in all dimensions of the Bryant test increased during the pandemic. However, in contrast to the article by Forte et al. [10], which states that “Boys were significantly less likely than girls to report all measured emotional reactions”, boys showed a more significant increase than girls in the physical aggression dimension.

Regardless of gender, those who previously had daily contact with horses (ES) and spent more time in nature had higher aggression scores. This can be explained by the fact that due to the confinement, fewer opportunities existed for riding; additionally, being outdoors meant a greater change in our everyday lives than for those who were less used to it.

The contradiction can almost be interpreted as a “withdrawal symptom”; in the ES group, before the pandemic, generally fewer behavioral problems occurred, and children were more prosocial than those who do not deal with animals and are not in contact with them [37].

Being able to spend time with pets—given that it was possible to do so under the restrictions and people even had more time for that—had a positive effect on aggression levels. Mueller et al. found the same phenomenon in their research; namely that adolescents spent more time with their pets during the pandemic and often entered into interactions with them as a strategy for coping with stress. They found, however, that a buffering effect of companion animals on adolescents’ loneliness was not proved by the overall results, and so suggest that there are complexities in the relationships between pet ownership, attachment, loneliness, and coping with stress [28].

According to Ratschen et al. [29], the bonding between humans and animals can be related to the mental health of animal owners and the strength of this in the dimensions of emotional closeness or intimacy appears to be independent of animal species. The same study also found that owning a pet attenuated some of the adverse psychological effects of the COVID-19 siege. This is also supported by the fact that, in our study, the increase in aggression levels during the pandemic appeared to be mainly caused by the lack of contact with peers, as well as the decrease in time spent outdoors and in physical activity.

This indirectly confirms the positive effects of animal-assisted activities and contact with animals in preventing negative psychological reactions to loneliness, even during pandemic periods.

The research also points out that those with a stronger sense of coherence, especially in terms of manageability and meaningfulness, perceived the negative effects of the pandemic as relatively less bad, and were less likely to show symptoms of aggression.

During the pandemic, the boys’ sense of coherence was weaker compared to the sample that was taken in 2018, before the pandemic. This overlaps with observations from other studies that in acute stress situations, the sense of coherence temporarily weakens [46].

Our results showed that mainly regarding manageability and meaningfulness, those with a stronger sense of coherence perceived the negative effects of the pandemic as less bad; furthermore, in their case, symptoms indicating aggression also occurred less frequently. This confirms the research results that support a significant influence of the sense of coherence on the health behavior of young people [19].

## 5. Conclusions

The research results indicate that in an extraordinary situation—such as the curfew due to the pandemic—negative effects, including the tendency for aggression, are significantly influenced by the strength of an individual’s sense of coherence. Young people with a stronger sense of coherence manifested a lower tendency toward aggression, which suggests that they adapt more easily to unusual situations. It follows that developing a salutogenic sense of coherence constitutes an important task for education, and thus for schools.

The research also draws attention to how spending time in nature, dealing with animals, and increasing one’s physical activity directly, but also indirectly, through the sense of coherence, affects the tendency toward aggression caused by the curfew “siege” during the pandemic. Therefore, the narrower and wider social environment must strive to shift these negative factors in a positive direction.

One hopes the pandemic will not appear again; however, unexpected stress situations may occur in the future, and thus young people must be prepared to duel them.

## Figures and Tables

**Figure 1 ijerph-20-00769-f001:**
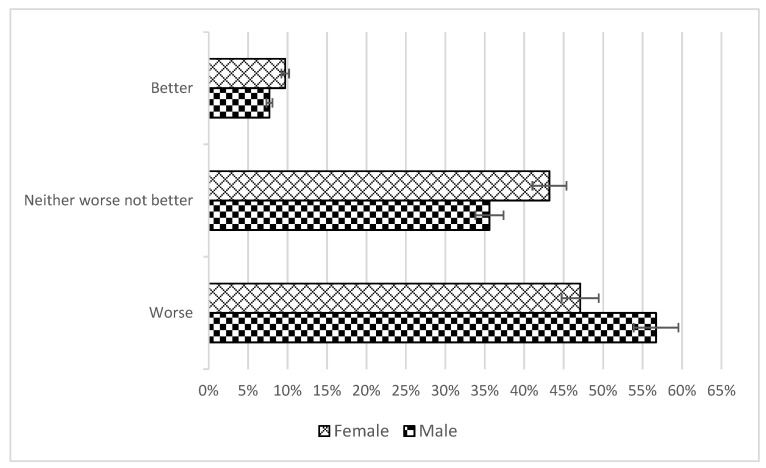
Distribution by gender of responses to the question “Overall, how would you rate your life under curfew compared to before the epidemic?”.

**Table 1 ijerph-20-00769-t001:** Characteristics of the study population.

		Research I		Research II		*p* *
		Mean	Std.	Mean	Std.	
Age		16.2	1.1	(16.0)	1.0	0.142
		Count	Col N%	Count	Col N%	*p* **
Gender	Male	152	29.0%	104	25.2%	0.206
	Female	373	71.0%	308	74.8%	
Place of residence	City	263	50.40%	198	48.10%	0.778
	Village	210	40.20%	173	42.00%	
	Farm	49	9.40%	41	10.00%	
Type of education	OS	193	36.8%	166	40.3%	0.270
	ES	332	63.2%	246	59.7%	

* *t*-test; ** Chi2-test.

**Table 2 ijerph-20-00769-t002:** Characteristics of the main outcome variables in the samples by gender.

	Research I (R.I.)			Research II (R.II.)			Male	Female
	Male	Female	Male vs. Female *	Male	Female	Male vs. Female *	R.I. vs. R.II *	R.I. vs. R.II *
	>Median	>Median	*p*	>Median	>Median	*p*	*p*	*p*
Comprehensibility	57.2%	43.2%	0.024	54.8%	41.9%	0.022	0.002	ns
Manageability	47.4%	52.5%	ns	35.6%	46.8%	ns	0.049	ns
Meaningfulness	50.7%	40.2%	ns	39.4%	54.5%	0.024	0.002	ns
Physical aggression	42.1%	51.7%	ns	60.6%	60.7%	ns	0.012	0.019
Anger	60.5%	65.7%	ns	73.1%	67.5%	ns	0.038	ns
Hostility	55.9%	55.5%	ns	72.1%	55.8%	0.003	0.027	ns
Verbal aggression	56.6%	55.8%	ns	69.2%	53.2%	0.004	0.049	ns

* Bonferroni correction applied for pairwise comparison.

**Table 3 ijerph-20-00769-t003:** Statistical characteristics of the samples by gender in SOC13 and Bryant12.

		SOC_13				Bryant_12			
		Count	Mean	95.0% Lower CL for Mean	95.0% Upper CL for Mean	Count	Mean	95.0% Lower CL for Mean	95.0% Upper CL for Mean
R.I.	Male	152	55.0	53.4	56.7	152	25.3 *	23.6	27.0
	Female	373	53.7	52.6	54.7	373	26.8	25.9	27.8
R.II.	Male	104	55.3	52.8	57.8	104	28.5 *	27.1	29.9
	Female	308	53.4	52.0	54.8	308	27.4	26.5	28.4

* For boys, according to the independent samples *t*-test, the difference in the means of the Bryant_12 test between surveys I and II is significant, *p* = 0.008.

**Table 4 ijerph-20-00769-t004:** Standardized regression coefficients and significance levels of the factors and independent variables assumed to influence the perception of coherence based on a multiple linear regression model.

SOC						
	Research I			Research II		
	Beta *	t	Sig.	Beta *	t	Sig.
Age (year) 14–16 (=0) vs. 18–18 (=1)	0.023	0.511	0.610	0.136	1.775	0.069
Gender Male (=0) vs. Female (=1)	−0.053	−1.227	0.026	−0.060	−1.180	0.120
Group OS (=0) ES (=1)	0.019	0.415	0.048	0.048	−1.335	0.183

* Standardized Coefficients.

**Table 5 ijerph-20-00769-t005:** Standardized regression coefficients and significance levels of the factors and independent variables estimated to influence aggression susceptibility based on a multiple linear regression model.

Bryant						
	Research I			Research II		
	Beta *	t	Sig.	Beta *	t	Sig.
Age (year) 14–16 (=0) vs. 18–18 (=1)	0.085	1.892	0.059	−0.058	−1.178	0.240
Gender Male (=0) vs. Female (=1)	−0.040	−0.865	0.387	0.066	1.295	0.196
Group OS (=0) ES (=1)	0.009	0.197	0.844	0.113	2.218	0.027

* Standardized Coefficients.

**Table 6 ijerph-20-00769-t006:** Standardized regression coefficients and significance levels of the hypothesized determinants of aggression tested by a multiple linear regression model.

	Research I			Research II		
	Beta	t	Sig.	Beta	t	Sig.
Age (year) 14–16 (=0) vs. 18–18 (=1)	0.074	1.831	0.068	0.013	0.329	0.743
Gender Male (=0) vs. Female (=1)	−0.050	−1.200	0.231	0.037	0.891	0.373
Group OS (=0) ES (=1)	−0.032	−0.749	0.454	0.075	1.755	0.046
Comprehensibility	−0.062	−1.360	0.174	−0.119	−2.020	0.044
Manageability	−0.327	−6.888	<0.001	−0.363	−6.31	<0.001
Meaningfulness	−0.169	−3.980	<0.001	−0.208	−4.561	<0.001

**Table 7 ijerph-20-00769-t007:** Analysis of the factors influencing the propensity to be aggressive under curfew, using a linear regression model.

Independent Variables	Research II		
	Beta	t	Sig.
Age (year) 14–16 (=0) vs. 18–18 (=1)	−0.033	−0.663	0.508
Gender male (=0) vs. female (=1)	0.046	0.897	0.370
Place of residence Village (=0) vs. City (=1)	0.092	1.875	0.050
Time spent outdoors no change (=0) vs. decreased (=1)	0.140	2.132	0.026
Physical activity no change (=0) vs. decreased (=1)	0.079	−1.787	0.038
Missed classmates/friends? no (=0) vs. yes (=1)	0.140	−2.398	0.017
Time spent dealing with animals no change (=0) vs. decreased (=1)	0.186	−3.574	<0.001
Overall time spent on the Internet no change (=0) vs. decreased (=1)	0.197	3.918	<0.001

Dependent Variable: Bryant 12 test.

## Data Availability

Not applicable.

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
