# Peer review of "Sense of Coherence and Self-Rated Aggression of Adolescents during the First Wave of the COVID-19 Pandemic, with a Focus on the Effects of Animal Assisted Activities"

_ijerph, 2022, doi:10.3390/ijerph20010769_

Round 1
Reviewer 1 Report
The topic is interesting and important. But main problem is in the writing, particularly in the results and discussion sections, which are-
1. repeated presentation of table e.g., table 2 and 4. Too much tables. Please present the data briefly.
2. data have not been properly discussed. Please discuss data on the basis of their importance
Reviewer 2 Report
This paper still needs further revision.
A discussion of the definition of sense of coherence could be added to the literature review.
The relationship between human and horse or horsemanship should be discussed further in the literature review, which could help this research have a more in-depth view in the discussion section instead of simply describing the results.
Reviewer 3 Report
The topic of the research is interesting. The authors show how the adverse effects induced by the Covid-19 Pandemic, especially on aggressiveness, can be partially counterbalanced by physical contact with a pet. The manuscript can be improved by including some references and revising the results section. The authors inserted a lot of tables but only one figure I suggest trying to transform some of the data shown in tables into graphs in order to increase the readability of the manuscript
line 45 It is necessary to add some references.
lines 161-162 Do you have some references in order to justify that the internal reliability indexed with Cronbach's a for the two surveys was within the acceptable scope?
line 246 and line 260 I suppose that the authors erroneously numbered the table and thus in the manuscript there were two table2. the table at line 246 should be the number two and the other at line 260 should be table 3.
line 258 please correct the p value
line correct the Pearson's rho value cited in the text that was different than the value shown in the table.
line 270 the table should be the number 4, correct also in the text. Correct also all the number of the other tables.
lines 386-387 I don't understand this sentence. In which phase of the research did the authors find this result? it is the opposite results presented in lines 388-390. Please clarify
Round 2
Reviewer 1 Report
Quality of the manuscript has much improved. Please discuss the table a bit more. It has just a title and two sentences. If not possible, move to the supplementary data section.
Author Response
Dear Reviewer 1,
we are very thankful for your comments and suggestions:
„Quality of the manuscript has much improved. Please discuss the table a bit more. It has just a title and two sentences. If not possible, move to the supplementary data section.”
R: We have corrected the Table 7. and the discuss.
Yours sincerely, on behalf of the authors,
Ákos Levente Tóth